# Association of VDR Polymorphisms with Muscle Mass Development in Elite Young Soccer Players: A Pilot Study

**DOI:** 10.3390/sports12090253

**Published:** 2024-09-13

**Authors:** Laura Flore, Renato Robledo, Laura Dettori, Marco Scorcu, Paolo Francalacci, Filippo Tocco, Myosotis Massidda, Carla Maria Calò

**Affiliations:** 1Department of Life and Environmental Sciences, University of Cagliari, 09042 Monserrato, CA, Italy; laura.flore@unica.it (L.F.); laura.dettori@unica.it (L.D.); paolo.francalacci@unica.it (P.F.); cmcalo@unica.it (C.M.C.); 2Department of Biomedical Sciences, University of Cagliari, 09042 Monserrato, CA, Italy; 3Cagliari Calcio SPA, Loc. Sa Ruina, 09032 Assemini, CA, Italy; mscorcu@tiscali.it; 4Department of Medical Sciences and Public Health, University of Cagliari, 09042 Monserrato, CA, Italy; filippo.tocco@tiscali.it (F.T.); myosotis.massidda@unica.it (M.M.)

**Keywords:** vitamin D receptor, SNPs, genotype, muscle mass

## Abstract

The vitamin D receptor (*VDR*) is an important candidate gene in musculoskeletal phenotypes. Polymorphisms in the *VDR* have been previously associated with several pathologies and muscular strength in athletes and elderly people; however, the literature reported contradictory results. The object of this research was to verify the association between the most studied *VDR* variants (rs2228570, rs7975232, and rs1544410) and the increase in muscle mass in elite young soccer players. A sample of 55 soccer players (15–18 years old) from a professional team were selected for this study. DNA was extracted by the salting-out method, and polymorphisms were genotyped by PCR-RFLP, followed by 2% agarose gel electrophoresis. To test the effect of the three SNPs (single nucleotide polymorphisms), a logistic regression analysis was applied. The body composition determination was carried out through the skinfold thickness method, and the muscular area of the arm and lower limb were calculated using the Frisancho formula. All three polymorphisms met the Hardy–Weinberg equilibrium (*p* > 0.05) and their frequencies fell within the worldwide variability. A significant correlation between rs1544410 and the increase in calf muscle mass was observed. Individuals carrying the A allele showed higher calf muscular mass than those carrying the G allele (*p* = 0.034). Moreover, a haplotype analysis applied to the two SNPs in linkage disequilibrium (rs7975232 and rs1544410) showed that the AG haplotype appeared negatively correlated to the calf muscle area. In conclusion, we confirm an association between *VDR* polymorphisms and muscular mass that could encourage the genetic screening of the *VDR* gene to identify a potential risk of injury and for individual nutritional interventions.

## 1. Introduction

Vitamin D, also called calciferol, is a fat-soluble molecule belonging to the family of steroid hormones and plays an important role in the regulation of calcium metabolism, bone [1], and muscle development. Vitamin D is involved in the mechanism of muscle contraction [2] and a strong relation between vitamin D level and sarcopenia has been demonstrated [3]. Moreover, it has been shown that sufficient levels of vitamin D enable the maintenance and development of athletic performance; consequently, reduced levels of the vitamin inhibit the muscle relaxation required after training and increase muscle pain. This leads to a major risk of injury and stress fractures, causing the loss of muscle power and reducing bone mineral density [4].

Vitamin D can either be consumed with the diet or synthesized by the skin: UVB (Ultraviolet B) rays, by penetrating the epidermis, converting 7-dehydrocholesterol (7DHC) into vitamin D_3_ (cholecalciferol).

The *VDR* gene is located on human chromosome 12q13.11 and comprises 11 exons. Exons 2 and 3 are involved in DNA binding, whereas exons 4 and 9 are involved in ligand binding [5]. Among the more than one hundred SNPs (single-nucleotide polymorphisms) present in the *VDR* gene, the most common and studied are rs228570 (FokI), located in exon 2, and rs1544410 (BsmI) and rs7975232 (ApaI), both located in intron 8. It has been suggested that these SNPs can determine the different responses to strength training [6], muscle injury [7], and hypertrophic response to resistance exercise in humans [8], but the results are still controversial [9].

The FokI polymorphism is located at the 5′ end of exon 2, corresponding to the start codon. This mutation involves the replacement of a guanine with an adenine. The A allele was found to be correlated to the maintenance of bone homeostasis [10,11,12]. The same allele has been significantly associated with total fat-free mass (FFM) in elderly, appendicular FFM, and relative appendicular FFM, demonstrating a correlation with sarcopenia [13]; accordingly, males carrying the GG genotype had significantly less FFM and appendicular fat-free mass and a lower skeletal muscle index (SMI) compared to A allele carriers, and a two-fold higher risk of being sarcopenic when compared to carriers of the A allele.

Other studies demonstrated an association between the FokI polymorphism and muscle phenotypes with male GG homozygotes showing significantly lower values of knee extensor (KE) [14] and handgrip (HG) strength [15] than A allele carriers. On the contrary, a study conducted by Gussago et al. [16] found AA individuals displaying significantly greater HG strength than G carriers.

BsmI, located at the 3′ end of the *VDR* gene, shows a change from A to G. The functional implications of this SNP include the possibility of altering splice sites involved in mRNA transcription or affecting the intronic regulatory elements of the *VDR* gene [17].

Its relationship with risk of osteoporosis has been widely debated with conflicting results: a meta-analysis conducted on 26 studies [18] reported the possible association between BsmI polymorphism and the risk of bone fracture, suggesting a protective role of GG for osteoporosis risk. A different meta-analysis conducted on 41 studies [19] did not confirm such an association.

Studies on the association between BsmI polymorphism and muscle mass and strength also highlighted contrasting results. While previous studies have denied any association between rs1544410 BsmI polymorphism and muscle mass phenotype [16], one study displayed higher quadriceps and grip strength for the elderly with the G allele [15]. In contrast, an association between AA genotypes and higher hamstring strength was demonstrated in a sample of elderly women [20,21], suggesting a higher knee extensor strength for AA genotypes when compared with GG and AG genotypes.

ApaI, also located at the 3′ end, shows a change from A to C. The *VDR* ApaI is associated with increased bone mass concomitant with higher calcium intake [22], and it seems to influence the severity of musculoskeletal injuries in professional football players [8]. A recent meta-analysis [23] highlighted an increased risk for osteoporotic fracture for AA genotypes in the European population but not in the overall sample that included individuals from the Americas (northern, central, and southern).

Several studies are based on the analysis of the association between *VDR* polymorphisms and bone mineral density (BMD) [24]. Limited research focused on the association with muscular mass and strength and has led to contrasting results. Research by Iki [25] negated any difference in developing muscular strength for the different genotypes. On the contrary, an association between genotypes and hand grip strength has been suggested in research on the elderly in Taiwan [26]. Moreover, a study by Wang [27] found lower knee and elbow concentric or eccentric peak torque among individuals carrying the ApaI A allele.

Several factors may account for the conflicting results in association studies, including different study designs, different methodologies used, and, above all, different population ethnicities.

As mentioned above, few studies focused on the relation between FFM and *VDR* variants, and even less dealing with young athletes.

The aim of the present research is to determine the association between the most studied *VDR* variants (FokI rs2228570, ApaI rs7975232, and BsmI rs1544410) and the increase in lean and muscle mass in young soccer players.

This pilot study is part of a broader project aimed at understanding the effects of VDR variants and its haplotype on human muscle and strength-related phenotypes.

The results may provide a better understanding of the physiology of muscle performance among young athletes and may be useful in developing optimal training programs.

## 2. Materials and Methods

### 2.1. Sampling

A total of 55 male elite soccer players, 15 to 18 years old (mean: 16.65 ± 1.55 years), were selected for this study. All players included in the study were from the same professional team and participated in the Official National Football Championship (Primavera, Allievi, and Giovanissimi). The inclusion criteria were as follows: Italian nationality, born in Italy, resident in Italy for at least 3 generations (self-reported), male sex, playing soccer for at least 5 years, and playing at Cagliari Calcio teams for at least 2 years.

All study participants read and signed an informed consent form, which was signed by the parents of underage athletes. This study was approved by the Ethics Committee of Azienda Ospedaliera Universitaria (AOU) of Cagliari University (Italy).

### 2.2. DNA Analyses

A buccal swab was taken by using a sterile cytobrush from each participant, and DNA was extracted using the salting-out method. Following lysis with proteinase K and sodium dodecyl sulfate (SDS), a saturated NaCl solution (5M) was used to precipitate the proteins. Centrifugation at 8000 rpm for 10′ removed the pellet. DNA was then precipitated by adding 100% isopropanol and isolated by centrifugation.

Concentration and quality of the extracted DNA were measured using the NanoDrop spectrophotometer (Thermo Fisher Scientific). DNA genotyping was performed by the RFLP-PCR method using the following primers for the three polymorphisms:ApaI: forward 5′-CAGAGCATGGACAGGGAGCAAG-3′; reverse 5′-CAACTCCTCATGGCTGAGGTCT-3′;BsmI: forward 5′-CAACAAGACTACAAGTACCGCGTCAGTGA-3′; reverse 5′-AACCAGCGGAAGAGGTCAAGGG-3′;FokI: forward 5′-GATGCCAGCTGGCCCTGGCACTG-3′; reverse 5′-ATGGAAACACCTTGCTTCTTCTCCCTC-3′.

Each reaction mix (total 25 µL) for all three polymorphisms included 2 µL of DNA, 0.2 µL of the forward and reverse primers, 10.6 µL of water, and 12 µL of NZYSupreme qPCR Green Master Mix 2x (Lisbon, Portugal,). Amplicons were digested with appropriate restriction enzymes and subjected to the electrophoresis technique, with a 2% agarose gel for ApaI and BsmI and a 10% polyacrylamide gel for FokI, observing the following DNA fragments: for ApaI (A: 740 bp, C: 515 + 225 bp); for BsmI (A: 825 bp, G: 650 + 175 bp); for FokI (G: 272 bp, A: 198 + 74 bp).

### 2.3. Body Composition Determination

For each subject, height and weight were measured according to standard procedure [28].

Body composition parameters were calculated using the skinfold equations. Two different formulas were used to calculate the percentage of fat mass (FM) and fat-free mass (FFM): Slaughter’s formula [29], designed specifically for boys (up to age 18) using measurements of the tricipital skinfold and subscapular skinfold; and Reilly’s formula [30], specifically validated for young soccer players (age 16.67 ± 0.5 years) by [31], which uses measurements of the thigh, abdominal, triceps, and calf skinfold. The muscle area of the arm (AMA), calf (CMA), and thigh (TMA) were calculated according to the Frisancho formula [32]: [Arm circumference − (tricep skinfolds × π)^2]/4π.

The AMA can be used as indicators of muscle mass and for predicting FM, and previous results demonstrated a good correlation with DEXA (Dual Energy X-ray Absorptiometry) results [33] and segmental BIA (Bioelectrical Impedance Analysis) [34].

Analogously, CMA and TMA were calculated using calf and thigh circumferences and calf and front thigh skinfolds. Both CMA and TMA values have been significant correlated with BIA analysis values [35].

For the AMA, correspondence to proper percentile for each participant was also verified.

### 2.4. Statistical Analyses

Allele and genotype frequencies, as well as Hardy–Weinberg, were calculated with the Genepop software platform (ver.4.4.3, Montpellier, ISEM, France).

A one-way ANOVA was used to evaluate the hypothesis of the equality of means among genotypes in relation to all body composition measures. Analyses were performed with the R 4.2.3 program (R Foundation for Statistical Computing, Vienna, Austria).

Genotypic association analysis with fat-free mass and muscle mass was performed with the SNPstats software platform (https://snpstats.net/ (accessed on 8 April 2024), Institut Català d’Oncologia, Barcelona, Spain).

Given the variability of the three polymorphisms across 1000 genomes, the population branch statistic (PBS), a summary statistic that exploits pairwise genetic differentiation (F_ST_) values among three populations, was calculated to check for evidence of natural selection within the *VDR* gene [36]. A population’s PBS value represents the amount of allele frequency change at a given locus in the history of this population. Loci with high PBS values have likely undergone a positive selection. The VCF (Variant Call Format) file of chromosome 12 was downloaded from the 1000 Genomes Database Phase 3. From the VCF file of the whole gene, only SNPs with a minimum allele frequency (MAF) ≥ 0.05 were selected, using Plink 1.9 (www.cog-genomics.org/plink/1.9/ (accessed on 9 April 2024) [37]); next, the F_ST_ index was calculated among three potential population pairs, namely YRI, CEU, and CHB, using the program VCFtools ver.3.0, https://vcftools.github.io/index.html (accessed on 9 April 2024) [38]; finally, the PBS was calculated and a two-dimensional graph created using R Studio (ver. 4.3.0, Boston, MA, USA) [39]. In order to obtain further information regarding the possible natural selection of the VDR gene, Tajima’s D was calculated using VCFtools ver3.0. Tajima’s D, a measure of nucleotide diversity, is used to compare an observed nucleotide diversity against the expected diversity under the assumption of selective neutrality and constant population size.

## 3. Results

Anthropometric characteristics of the sample are shown in Appendix A, while allele and genotype frequencies are shown in Table 1. The Hardy–Weinberg test revealed that all samples were in equilibrium for all loci (*p* > 0.05). The three SNPs showed a high worldwide genetic variability (1000 Genomes dataset), and allele frequencies of our sample fell within the worldwide range; ApaI and BsmI frequencies were also included within the European range (Table 1).

Considering the high variability in allele frequencies and the correlation of the gene with several diseases, we examined, through PBS, possible traces of natural selection using data from 1000 genomes. Some SNPs had PBS values near or equal to the limits of the distribution, but not the specific polymorphisms we focused on. They fell on both lines and had consistently low PBS values, suggesting the absence of selection (Figure 1).

The absence of selection was confirmed by Tajima’s D test: positive values were obtained for the three SNPs (2.486 for ApaI, 1.957 for BsmI, and 2.113 for FokI), indicating low levels of low- and high-frequency polymorphisms.

A one-way ANOVA test was used to assess any differences in the fat-free mass between the different genotypes. The data showed no significant differences between genotypes for any considered variables (*p* > 0.05, Table 2). It is noteworthy to mention a weak trend of increasing CMA values from CC to AA for ApaI (88.09, 94.27, and 95.72 for CC, AC, and AA, respectively).

The logistic regression analyses revealed a significant association between BsmI polymorphism and CMA under the dominant model: athletes carrying the A allele had a higher calf muscle mass than GG homozygous (Table 3). Moreover, although not statistically significant, we found higher values of CMA and AMA in individuals with the FokI AA genotype under the recessive model (97.32 vs. 93.49 and 54.22 vs. 49.71, respectively).

To test the combined effects of ApaI rs7975232 and BsmI rs1544410 SNPs, we applied a logistic regression analysis to the haplotypes. The linkage disequilibrium of these variants for European population was previously verified through the LDlink program [40], attesting a strong linkage between ApaI and BsmI. This result was confirmed in our sample by applying D statistics through the SNPstats program (https://www.snpstats.net/start.htm (accessed on 10 April 2024)).

A significant association was found between the CMA and AG haplotype (Table 4): the negative value of difference (95% CI) suggests a significantly low muscle mass for individuals carrying the AG haplotype.

In addition, despite not reaching the level of significance, the AG haplotype also showed a lower muscle mass for arm and thigh area (Appendix A).

## 4. Discussion

Vitamin D has been proven to influence muscle activity and performance [41]. The literature shows some evidence on the association between vitamin D receptor polymorphisms and muscle strength, but the issue is still under debate. Most of the literature deals with the relationship between *VDR* polymorphisms and decreasing muscle mass with aging or the analysis of muscular traits in young athletes [16].

In this study, we analyzed the association between *VDR* polymorphisms and muscle mass, in a sample of elite young soccer players from a professional Italian team, to determine the relationship between muscle mass and *VDR* variants.

A PBS analysis showed no evidence of selection for the three polymorphisms: their high level of variability is likely due to random evolutive factors, such as genetic drift.

Previous research has focused on the correlation between *VDR* polymorphisms, muscle mass, and sarcopenia in elderly [13,42] and found that individuals with the FokI GG genotype were more likely to have a lower muscle mass and to develop sarcopenia than AA and AG genotypes. Our sampling includes only young athletes, and therefore, we could not detect any correlation with sarcopenia; however, our data are in good agreement with previous reports, suggesting that FokI rs2228570 can lead to different responses to training by increasing the muscle size and strength [19] even in young soccer players. Indeed, the logistic regression highlighted higher values of CMA in individuals carrying the A allele, suggesting a possible correlation between FokI rs2228570 and muscle mass. The molecular and physiological basis for this association is still uncertain, but previous studies suggested that the G allele produces an isoform with a decreased ability to induce transcription compared to the isoform coded by the A allele variant [43], and, consequently, a reduced muscle mass.

The important role of FokI rs2228570 in soccer players’ performance was also suggested in a previous study [44] showing an increase in homozygous AA individuals among young soccer players than in the control group. This result could suggest that the analysis of the FokI polymorphism could help select young athletes who possess the most favorable genetic potential to succeed in soccer.

Moreover, our study suggested a significant correlation between BsmI polymorphism rs1544410 and the development of calf muscle mass. The dominant model shows a statistically significant increase in CMA among individuals carrying the A allele. This result was never found in previous research; indeed the genotype AA was only correlated with hamstring strength, although the authors did not find any correlation with muscle mass [20,21].

Finally, ApaI rs7975232 did not show any statistical association with muscle mass, even though a weak trend versus a higher increase in CMA was evident in CC carriers with respect to AA carriers. To the best of our knowledge, only one study has previously found similar results: a research carried out on the elderly in Taiwan showed an association between hand grip strength and homozygous CC females [26].

The role of rs7975232 ApaI and rs1544410 BsmI SNPs in VDR activity is not fully understood, but it has been hypothesized that despite being located in a non-translated region of the gene, these two SNPs could have a role in mRNA stability due to their proximity to the poly-A site [45].

To verify the combined effect of the two SNPs in the linkage disequilibrium on muscle mass, we analyzed the correlation between haplotype and muscle mass, and, to our knowledge, this is the first time that this approach was used. The AG haplotype appeared negatively correlated with CMA. The data are particularly useful for its possible application in the field of sports science, since individuals carrying the AG haplotype, given the same amount of training, seem to develop less calf muscle mass. It would be interesting to be able to confirm whether this finding also determines earlier sarcopenia in the elderly.

## 5. Conclusions

This study supports the importance of studying the simultaneous effects of different allelic variants to better define the relationship between genotypes and phenotypes.

To the best of our knowledge, this is the first study dealing with *VDR* polymorphisms and mass development in young Italian soccer players. The results could add precious information for predicting athletic performance and could help in developing individual training on the basis of *VDR* variants.

Our data suggest an association between FokI rs2228570 and muscular mass, with A carriers having a higher CMA as compared to their counterparts. In addition, the association found between both BsmI rs1544410 and the AG haplotype (ApaI rs7975232 and BsmI rs1544410) with an increase in calf muscle mass suggests the potential use of genetic screening for the *VDR* gene in athletes or in elderly individuals to identify a potential risk of injury and/or for individual nutritional interventions, such as vitamin D supplementation, as suggested by previous research [41,46].

Although the sample size is not large, we believe that the highly homogeneous population tested may be considered adequate for a pilot study. However, an analysis of a much larger sample size is required to validate our conclusions.

## Figures and Tables

**Figure 1 sports-12-00253-f001:**
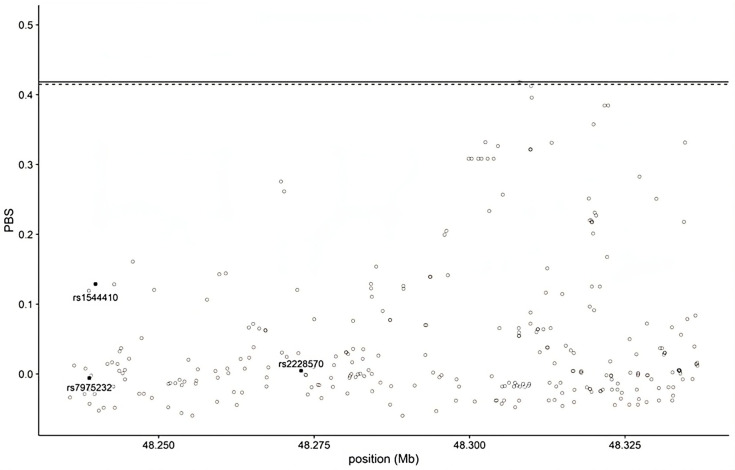
Empirical distribution of PBS values for the *VDR* gene region. Dashed and horizontal lines represent the 99.5th and 99.9th percentiles, respectively. SNPs under scrutiny are indicated with a square. In the Y axis, PBS values are reported, while in the X axis, the nucleotide positions along the chromosome are indicated.

**Table 1 sports-12-00253-t001:** Allele and genotype frequencies. Values are indicated in percentages, while absolute values are shown in parentheses.

***VDR* ApaI rs7975232 (N = 55)**				
A	C	AA	AC	CC
58.18	41.82	30.91 (17)	54.54 (30)	14.55 (8)
***VDR* BsmI rs1544410 (N = 54)**				
G	A	GG	AG	AA
55.56	44.44	35.18 (19)	40.74 (22)	24.08 (13)
***VDR* FokI rs2228570 (N = 54)**				
A	G	AA	AG	GG
25.93	74.07	3.71 (2)	44.44 (24)	51.85 (28)

N = sample size.

**Table 2 sports-12-00253-t002:** *p* values calculated for ANOVA test.

	ApaI		BsmI		FokI	
	F Value	*p*	F Value	*p*	F Value	*p*
FFM Slaughter	0.997484	0.375752	0.105224	0.900317	1.016038	0.369237
FFM Reilly	0.538281	0.586968	0.046829	0.954292	0.290343	0.749235
AMA	1.022569	0.366788	0.08582	0.917892	1.449376	0.244221
CMA	1.132188	0.330143	2.263845	0.1143	0.100998	0.904115
TMA	0.536284	0.588118	1.382144	0.260282	0.536284	0.588118

FFM: fat-free mass; AMA: arm muscle area; CMA: calf muscle area; TMA: tight muscle area; F value: the test statistic from the F test; *p*: the *p*-value of the F statistic.

**Table 3 sports-12-00253-t003:** BsmI association with AMA, CMA, and TMA.

Model	Genotype AMA	N	Response Means (s.e.)	Difference (95% CI)	*p*-Value	AIC	BIC
Codominant	GG	19	48.96 (2.32)	0	0.78	386	396
	AG	22	49.93 (1.83)	1.46 (−3.59–6.51)			
	AA	13	51.95 (1.92)	1.85 (−3.99–7.69)			
Dominant	GG	19	48.96 (2.32)	0	0.49	384	392
	AG–AA	35	50.68 (1.34)	1.60 (−2.94–6.14)			
Recessive	GG–AG	41	49.48 (1.44)	0	0.68	385	393
	AA	13	51.95 (1.92)	1.09 (−4.08–6.26)			
Overdominant	GG–AA	32	50.18 (1.58)	0	0.75	385	393
	AG	22	49.93 (1.83)	0.73 (−3.74–5.20)			
Log-additive				0.97 (−1.89–3.84)	0.51	385	392
	**Genotype CMA**						
Codominant	GG	19	89.04 (2.58)	0	0.11	420	430
	AG	22	95.37 (2.93)	6.93 (0.03–13.82)			
	AA	13	97.18 (2.34)	6.79 (−1.18–14.77)			
Dominant	GG	19	89.04 (2.58)	0	0.034	418	426
	AG–AA	35	96.05 (2.02)	**6.88 (0.68–13.08)**			
Recessive	GG–AG	41	92.44 (2.02)	0	0.4	422	430
	AA	13	97.18 (2.34)	3.16 (−4.15–10.46)			
Overdominant	GG–AA	32	92.34 (1.92)	0	0.19	421	429
	AG	22	95.37 (2.93)	4.26 (−1.99–10.52)			
Log-additive				3.71 (−0.25–7.67)	0.072	419	427
	**Genotype TMA**						
Codominant	GG	19	188.85 (6.23)	0	0.15	508	518
	AG	22	202.72 (6.18)	15.21 (−0.32–30.74)			
	AA	13	196.35 (5.79)	4.42 (−13.54–22.38)			
Dominant	GG	19	188.85 (6.23)	0	0.13	507	515
	AG–AA	35	200.35 (4.42)	11.23 (−2.93–25.39)			
Recessive	GG–AG	41	196.29 (4.48)	0	0.67	510	518
	AA	13	196.35 (5.79)	−3.56 (−19.98–12.86)			
Overdominant	GG–AA	32	191.9 (4.37)	0	0.06	506	514
	AG	22	202.72 (6.18)	13.48 (−0.26–27.21)			
Log-additive				3.37 (−5.73–12.46)	0.47	509	517

FFM: fat-free mass; AMA: arm muscle area; CMA: calf muscle area; TMA: tight muscle area; N: sample size; s.e.: standard error; CI: confidence interval; AIC: Akaike information criterion; BIC: Bayesian information criterion.

**Table 4 sports-12-00253-t004:** Haplotype association with response with CMA.

ApaI	BsmI	Frequency	Difference (95% CI)	*p*-Value
A	A	0.4138	0.00	---
C	G	0.3807	−3.68 (−8.74–1.39)	0.16
A	G	0.168	**−6.49 (−12.59–−0.4)**	**0.04**
C	A	0.0375	−11.09 (−27.62–5.45)	0.19

## Data Availability

Data are available upon request.

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
