# Peer review of "Association of VDR Polymorphisms with Muscle Mass Development in Elite Young Soccer Players: A Pilot Study"

_sports, 2024, doi:10.3390/sports12090253_

Round 1

Reviewer 1 Report

Comments and Suggestions for Authors

Intro – overall ok but need to make it shorter (some components might fit better in the discussion), and specifically link the previous research to the gap of knowledge leading to the purpose of the study.

Line 42: change “assumed” with “consumed” or “ingested”.

Line 42: add full name for UVB then use abbreviation.

From line 52 until line 61 – merge the small paragraphs into one paragraph.

Line 74 – add author’s last name then et al. before [17].

Line 71 until line 80 – merge the small paragraphs into one paragraph.

Line 82-84 – add reference(s).

Line 91: “While several research” – change to something like “While previous studies have”

Line 104-105 – add reference(s).

Line 112: change “verify” to “determine”.

Methods

Section 2.2 Dan Analysis: merge the small paragraphs into one paragraph.

Line 155-156: need to add more information about the calculation for muscle area for arm, calf, and thigh as well as add references showing that estimations of muscle area using the equation have been done before – preferably with confirmation from DEXA or MRI.

Results

Need to add table with subject characteristics – height, weight, BMI, muscle weights, etc.; also, it would be good to add the positions for the soccer players (goalkeeper, defender, midfielder, forward).

Line 189” change “focus” to “focused”.

Line 205: change “developed” to “had”.  Developed seems that athletes were monitored over time, which is not the case of the present study.

Table 3 should also include the results of BsmI with AMA and TMA.  These can be used as reference values for future studies.

Line 218: change “has been” to “was”. 

Discussion

Line 231-232:  the study was conducted to determine the relationship between muscle mass and VDR variants, and not “if the increase of muscular mass can be influenced by VDR variants.” – the way that this is currently written can be interpreted that VDR variants were “manipulated” which is not the case here.

Line 239-241: “A significant association has been found between CMA and haplotype AG (Table 4): the negative value of difference (95% CI) suggests significant low muscle mass for individuals carrying AG haplotype.” – how does the current data presented here agrees with the previous research study?  This statement is incorrect, unless the authors have results not presented in the results.

Line 241-244: which of the results shown support the statement from the authors?  From the results presented it seems that individuals with the AG – AA genotypes (BmsI - dominant model) had greater calf muscle mass. So how does this confirm the “role of FokI rs2228570 in developing muscular mass.”?

Line 251-254: the results do not “partially confirm” previous findings – previous findings did not report any correction with muscle mass.  This needs to be changed.

Line 266-267: “Haplotype AG appeared negatively correlated with muscular mass of arm, calf, and thigh,..” – these results are not significant, so this statement from the authors is incorrect.

Line 268: change word “datum” to “data”.

Conclusion

Line 276-278: unless some results were not presented clearly, there is not evidence from the current paper confirming “the role of FokI rs2228570 in developing muscular mass”.

Comments on the Quality of English Language

Some small corrections need to be made in the manuscript - please see comments and suggestions for authors.

Author Response

Intro – overall ok but need to make it shorter (some components might fit better in the discussion), and specifically link the previous research to the gap of knowledge leading to the purpose of the study.

Response: Thank you for taking the time to review our paper. We have revised the manuscript according to your precious suggestions.

Introduction was shortened, we eliminate lines 44-50 from the original manuscript, and lines 76 through 80 were moved to Discussion.

Line 42: change “assumed” with “consumed” or “ingested”.

Response: “assumed” was changed with “consumed”

Line 42: add full name for UVB then use abbreviation.

Response: full name for UVB was added

From line 52 until line 61 – merge the small paragraphs into one paragraph.

Response: Lines 52 through 61 were merged into a single paragraph

Line 74 – add author’s last name then et al. before [17].

Response: author’s last name was added

Line 71 until line 80 – merge the small paragraphs into one paragraph.

Response: Line 71 until line 80 was merge into a single paragraph

Line 82-84 – add reference(s). Response: reference was added

Line 91: “While several research” – change to something like “While previous studies have”

Response: “While several research” – was change to “While previous studies have”

Line 104-105 – add reference(s). Response:  Thank you for catching it. Reference was added

Line 112: change “verify” to “determine”. Response: “verify” was changed to “determine”

Methods

Section 2.2 Dan Analysis: merge the small paragraphs into one paragraph.

Response: DNA Analysis: the two small paragraphs were merged into a single paragraph

Line 155-156: need to add more information about the calculation for muscle area for arm, calf, and thigh as well as add references showing that estimations of muscle area using the equation have been done before – preferably with confirmation from DEXA or MRI.

Response: Line 155-156: Thank you for your suggestion. More information was added about the calculation for muscle area for arm, calf, and thigh, including formulae and references

Results

Need to add table with subject characteristics – height, weight, BMI, muscle weights, etc.; also, it would be good to add the positions for the soccer players (goalkeeper, defender, midfielder, forward).

Response: Thank you for your suggestion. We added a table with subject characteristics along with their position on the pitch. The table is available as supplementary material.

Line 189” change “focus” to “focused”. Response: “focus” was changed to “focused”

Line 205: change “developed” to “had”.  Developed seems that athletes were monitored over time, which is not the case of the present study. Response: We agree with this comment, developed” was changed to “had”  

Table 3 should also include the results of BsmI with AMA and TMA.  These can be used as reference values for future studies. Response: We agree with this comment: Table 3 now includes the results of BsmI also with AMA and TMA

Line 218: change “has been” to “was”. Response: “has been” was changed to “was”.

Discussion

Line 231-232:  the study was conducted to determine the relationship between muscle mass and VDR variants, and not “if the increase of muscular mass can be influenced by VDR variants.” – the way that this is currently written can be interpreted that VDR variants were “manipulated” which is not the case here. Response: Thank you pointing this out, the sentence was rephrased

Line 239-241: “A significant association has been found between CMA and haplotype AG (Table 4): the negative value of difference (95% CI) suggests significant low muscle mass for individuals carrying AG haplotype.” – how does the current data presented here agrees with the previous research study?  This statement is incorrect, unless the authors have results not presented in the results.

Response: We did not say that the data presented in table 4 agrees with the previous research study, however we verified and corrected our statement referring the concordance with previous studies. Moreover, for a better comprehension of the results we added table S2 with non-significant results for haplotype association with response with AMA and TMA (replacing data not shown with Table S2)

Line 241-244: which of the results shown support the statement from the authors?  From the results presented it seems that individuals with the AG – AA genotypes (BmsI - dominant model) had greater calf muscle mass. So how does this confirm the “role of FokI rs2228570 in developing muscular mass.”? Response: we thank the reviewer, the sentence was modified

Line 251-254: the results do not “partially confirm” previous findings – previous findings did not report any correction with muscle mass.  This needs to be changed. Response: We agree with the reviewer, the sentence was rephrased

Line 266-267: Haplotype AG appeared negatively correlated with muscular mass of arm, calf, and thigh,..” – these results are not significant, so this statement from the authors is incorrect. Response: Thank you for pointing this out, we agree, the sentence was rephrased

Line 268: change word “datum” to “data”. Response: the word “datum” was changed to “data”.

Conclusion: unless some results were not presented clearly, there is not evidence from the current paper confirming “the role of FokI rs2228570 in developing muscular mass”. We agree with the reviewer, at Line 276-278 we rephrased the sentence                   

Comments on the Quality of English Language

Some small corrections need to be made in the manuscript - please see comments and suggestions for authors.

Response: Thank you for the time you dedicated to review our manuscript. We follow all your precious suggestions.

Reviewer 2 Report

Comments and Suggestions for Authors

Reviewer Comments

This study is an intriguing pilot investigation exploring the association between VDR gene polymorphisms and muscle mass development in elite youth soccer players. It provides valuable insights into the relationship between VDR gene polymorphisms and muscle mass development in young athletes. Overall, the paper is well-structured and addresses an important topic. However, there are several areas that require improvement.

Introduction

The introduction effectively presents background information on the relationship between vitamin D and muscle development and function. However, it would be beneficial to elaborate on why this topic is particularly significant for young athletes.

The review of existing studies on the relationship between VDR gene polymorphisms and muscle phenotypes is comprehensive. Nevertheless, a more detailed explanation of the inconsistencies among some research findings is necessary.

Some technical terms and abbreviations lack explanation. For instance, it would be helpful to add brief explanations for terms such as 'PBS' and 'Tajima's D test'.

Methods

Please provide precise information about the software used for data processing (e.g., company name, country).

Results

Add explanations for abbreviations below the tables.

Figure 1 (PBS analysis) provides useful information, but the axis labels and legend should be clearer. Additionally, improving the resolution of the figure would be beneficial.

Discussion

Overall, the discussion section effectively summarizes and interprets the research findings.

It would be beneficial to briefly add more detailed information about the practical applicability of the research results.

Author Response

This study is an intriguing pilot investigation exploring the association between VDR gene polymorphisms and muscle mass development in elite youth soccer players. It provides valuable insights into the relationship between VDR gene polymorphisms and muscle mass development in young athletes. Overall, the paper is well-structured and addresses an important topic. However, there are several areas that require improvement.

 Response: Thank you for your positive comment. We have revised the manuscript according to your precious suggestions.

Introduction

The introduction effectively presents background information on the relationship between vitamin D and muscle development and function. However, it would be beneficial to elaborate on why this topic is particularly significant for young athletes.

Response: Thank you for your suggestion. A sentence was added explaining why the topic is particularly significant for young athletes

The review of existing studies on the relationship between VDR gene polymorphisms and muscle phenotypes is comprehensive. Nevertheless, a more detailed explanation of the inconsistencies among some research findings is necessary.

Response: Thank you for your comment. A sentence was added explaining the conflicting results from the literature in association studies

Some technical terms and abbreviations lack explanation. For instance, it would be helpful to add brief explanations for terms such as 'PBS' and 'Tajima's D test'.

Response: Thank you for pointing it out. We explained all technical terms and abbreviations

 Methods

Please provide precise information about the software used for data processing (e.g., company name, country).

Response: Thank you for your suggestion. We provided precise information about the software used for data processing

 Results

Add explanations for abbreviations below the tables. Response: Explanations for abbreviations below the tables are provided

Figure 1 (PBS analysis) provides useful information, but the axis labels and legend should be clearer. Additionally, improving the resolution of the figure would be beneficial.

Response. Thank you for your suggestion. We modified the legend of figure 1, adding information on the Y and X axis. We also improved the resolution of Figure 1

 Discussion

Overall, the discussion section effectively summarizes and interprets the research findings.  It would be beneficial to briefly add more detailed information about the practical applicability of the research results.

Response: Thank you for your comment. A sentence is added, in the Conclusions, about the practical applicability of the research results

Reviewer 3 Report

Comments and Suggestions for Authors

The manuscript intended for evaluation by the author team of Laura Flore et al. ‘ Association of VDR Polymorphisms with Muscle Mass Development in Elite Young Soccer Players: A Pilot Study ‘ is interesting and innovative. The article is well structured into sections and subsections.

However, I have some comments:

1.      PCR-RFLP method was used in the study, „rs” numbers should be used to determine polymorphisms in RT-PCR method.

2.      It is accepted that the names of genes should be written in italics, in fact in some places the authors use this spelling.

 3.      I suggest shortening the introduction, part of the content is actually already a discussion.

 4.      A limitation of the study is the small study group and lack of a control group. It is difficult to draw statistically significant associations on this basis.

 5.      The pilot study should present the assumptions of the main study and justify the pilot study carried out.

 6.      What are the inclusion and exclusion criteria on the basis of which this study group size was established?

 7.      Please describe the DNA isolation method.

 8.      Why are there N=54 and N=55 in Table 1?

Author Response

The manuscript intended for evaluation by the author team of Laura Flore et al. ‘ Association of VDR Polymorphisms with Muscle Mass Development in Elite Young Soccer Players: A Pilot Study ‘ is interesting and innovative. The article is well structured into sections and subsections.

 Response: Thank you for your positive comment. We have revised the manuscript according to your precious suggestions.

  1. PCR-RFLP method was used in the study, „rs” numbers should be used to determine polymorphisms in RT-PCR method. Response: We agree with the reviewer, but to facilitate the readers in comparing data from the literature, we utilized both „rs” numbers together with the enzyme nomenclature.
  2. It is accepted that the names of genes should be written in italics, in fact in some places the authors use this spelling. Response: Thank you for catching it. All gene names of genes are now written in italics
  3. I suggest shortening the introduction, part of the content is actually already a discussion. Response: We thank for your comment. According to reviewer suggestion, introduction was shortened, we eliminate lines 44-50 from the original manuscript, and lines 76 through 80 were moved to Discussion.
  4. A limitation of the study is the small study group and lack of a control group. It is difficult to draw statistically significant associations on this basis. Response: We agree that the small number of samples is a limitation of the study, however, considering also the homogeneity of the population (all the footballers played for the same professional team), we believe that the number of participants considered could be adequate for a pilot study.
  5. The pilot study should present the assumptions of the main study and justify the pilot study carried out.

Response: Thank you for your suggestion. We added two sentences, in the Introduction, presenting the assumptions of the main study and   justifying the pilot study carried out.

  1. What are the inclusion and exclusion criteria on the basis of which this study group size was established? Response: thank you for your comment. Inclusion criteria have been reported
  2. Please describe the DNA isolation method. Response: Thank you for your suggestion.  A brief description of DNA isolation method was added.
  3. Why are there N=54 and N=55 in Table 1? Response: The genotyping of one individual failed in two polymorphisms

Round 2

Reviewer 3 Report

Comments and Suggestions for Authors

I am satisfied with the authors' responses.